# Monocyte Activation and Ageing Biomarkers in the Development of Cardiovascular Ischaemic Events or Diabetes in People with HIV

**DOI:** 10.3390/microorganisms11071818

**Published:** 2023-07-16

**Authors:** Jose I. Bernardino, Belen Alejos, Javier Rodriguez-Centeno, Andrés Esteban-Cantos, Beatriz Mora-Rojas, Rocío Montejano, Rosa De Miguel, Marta Montero-Alonso, Oskar Ayerdi, Cristina Hernández-Gutierrez, Adriá Curran, Jose R. Arribas, Berta Rodés

**Affiliations:** 1Unidad de VIH, Hospital Universitario La Paz, IdiPAZ, 28046 Madrid, Spain; rocio.montejano@salud.madrid.org (R.M.); rosademiguelbuckley.med@gmail.com (R.D.M.); joser.arribas@salud.madrid.org (J.R.A.); 2CIBERINFECC, Instituto de Salud Carlos III, 28029 Madrid, Spain; javier.rodriguez@idipaz.es (J.R.-C.); andres.esteban@idipaz.es (A.E.-C.); beatriz_mora19@hotmail.com (B.M.-R.); berta.rodes@salud.madrid.org (B.R.); 3Centro Nacional de Epidemiología, Instituto de Salud Carlos III, 28029 Madrid, Spain; belen.alejos.ferreras@gmail.com; 4HIV/AIDS and Infectious Diseases Research Group, IdiPAZ, 28046 Madrid, Spain; 5Unidad de Enfermedades Infecciosas, Hospital Universitario y Politécnico La Fe, 46026 Valencia, Spain; martamontero72@gmail.com; 6Centro Sanitario Sandoval, Hospital Clínico San Carlos, IdiSSSC, 28010 Madrid, Spain; oskarayerdi@hotmail.com; 7Departamento de Medicina Interna, Hospital Universitario Príncipe de Asturias, Alcalá de Henares, 28805 Madrid, Spain; cris.hg.86@gmail.com; 8Departamento Enfermedades Infecciosas, Hospital Universitari Vall d’Hebron, VHIR, 08035 Barcelona, Spain; acurran@vhebron.net

**Keywords:** HIV infection, blood telomere length, epigenetic age acceleration, monocyte activation, cardiovascular disease, diabetes

## Abstract

We investigated whether blood telomere length (TL), epigenetic age acceleration (EAA), and soluble inflammatory monocyte cytokines are associated with cardiovascular events or diabetes (DM) in people living with HIV (PLHIV). This was a case–control study nested in the Spanish HIV/AIDS Cohort (CoRIS). Cases with myocardial infarction, stroke, sudden death, or diabetes after starting antiretroviral therapy were included with the available samples and controls matched for sex, age, tobacco use, pre-ART CD4 cell count, viral load, and sample time-point. TL (T/S ratio) was analysed by quantitative PCR and EAA with DNA methylation changes by next-generation sequencing using the Weidner formula. Conditional logistic regression was used to explore the association with cardiometabolic events. In total, 180 participants (94 cases (22 myocardial infarction/sudden death, 12 strokes, and 60 DM) and 94 controls) were included. Of these, 84% were male, median (IQR) age 46 years (40–56), 53% were current smokers, and 22% had CD4 count ≤ 200 cells/mm^3^ and a median (IQR) log viral load of 4.52 (3.77–5.09). TL and EAA were similar in the cases and controls. There were no significant associations between TL, EAA, and monocyte cytokines with cardiometabolic events. TL and EAA were mildly negatively correlated with sCD14 (rho = −0.23; *p* = 0.01) and CCL2/MCP-1 (rho = −0.17; *p* = 0.02). We found no associations between TL, EAA, and monocyte cytokines with cardiovascular events or diabetes. Further studies are needed to elucidate the clinical value of epigenetic biomarkers and TL in PLHIV.

## 1. Introduction

People living with HIV (PLHIV) have an excess risk for cardiovascular disease compared to the general population [1]. The risk for myocardial infarction in PLHIV is twofold higher. The incidence of stroke is 40% higher than the general population, even without traditional cardiovascular risk factors [2,3]. The contribution of cardiovascular disease to mortality is increasing in PLHIV. This directly results from ageing and reducing AIDS-related deaths [4]. Patients with type 2 diabetes, chronic kidney diseases, and HIV infection are characterised by premature vascular and accelerated biological ageing [5]. Apart from controlling traditional and non-traditional cardiovascular risk factors, there is an urgent need for novel biomarkers to identify those at a higher risk of cardiovascular disease.

Atherosclerosis is an inflammatory disease, and HIV infection is also characterised by chronic inflammation and an immune activation state that persists even years after virological suppression, contributing in part to a higher clinical risk for non-AIDS comorbidities [6]. Different studies have shown the predictive value of some inflammatory markers for death and non-AIDS-related conditions, including cardiovascular events [7,8]. Higher sCD163 levels are associated with a higher prevalence of coronary plaques and stenosis, and the quantity of inflammatory monocytes is predictive of cardiovascular events [9,10]. T-cell activation is a hallmark of atherosclerosis, and both innate and adaptive immune cells are critical players in plaque formation and progression [11,12].

Cellular and molecular processes such as telomere attrition and epigenetic changes such as DNA methylation are essential drivers of the ageing phenomenon [13]. Telomeres are TTAGGG repetitive sequences at the end of chromosomes that shorten with each cell division, and blood telomere length (TL) is a surrogate marker of chronological age. PLHIV have shorter telomeres than the general population [14]. A systematic review and meta-analysis have demonstrated an association between short TL and cardiometabolic events, including type 2 diabetes and several traditional cardiovascular risk factors associated with TL [15,16].

Cytosine–guanine dinucleotide (CpG) methylation is an epigenetic mechanism that modulates gene expression. DNA methylation is one of the most valuable biomarkers for predicting biological age [17]. Several methylation age calculators have been published and validated, accurately predicting chronological age in blood and other tissues [18,19]. Studies exploring epigenetic clocks in PLHIV demonstrate that this group consistently has a greater epigenetic age acceleration (EAA) than age- and sex-matched negative controls [20,21,22]. Moreover, HIV infection induces EAA and changes in the methylation pattern before antiretroviral treatment (ART) initiation and in long-term-treated individuals [23,24].

Both telomere length and DNA methylation age are independently associated with chronological age and mortality in general population cohorts [17,18,19]. A higher EAA in the general population is related to cardiovascular mortality [20].

Few studies have explored the association between these novel age biomarkers and cardiometabolic events in PLHIV. Our study aimed to elucidate whether TL and EAA are related to cardiovascular events in PLHIV.

## 2. Materials and Methods

### 2.1. Study Design and Participants

We designed a case–control study nested in the Spanish cohort of HIV-positive persons (CoRIS). CoRIS is an open, multicentre, and prospective cohort of adult PLHIV naïve to ART at entry, starting from 1 January 2004, in 46 centres from 13 autonomous regions in Spain. The subjects signed informed consent before admission, and ethical approval was granted from the Ethics Committee of Hospital Gregorio Marañón and each participating centre. For these analyses, the administrative censoring date was 30 November 2015. Briefly, CoRIS collects a minimum dataset for the cohort, including baseline and follow-up sociodemographic, immunological, and clinical data [25]. All centres provided data on the incident non-AIDS event, including non-AIDS-defining malignancies, cardiovascular, renal, liver, psychiatric, bone, and metabolic events fulfilling a structured event reporting form with the precise definition of each non-AIDS event, as previously reported [26]. This cohort is linked to a centralised repository (Spanish HIV Hospital Universitario Gregorio Marañón Biobank HIV HUGM BioBank) [27]. The centres have instructions to obtain a first blood sample before starting ART and annually after that whenever possible. All participants signed a specific, informed consent for BioBank sample storage. The authors did not have access to information that could identify individual participants.

We included participants in the cohort with available samples from BioBank. We requested the first available sample preferentially before ART initiation. All laboratory analyses were performed on batched samples between November 2019 and December 2020. Cases were defined as participants with a confirmed cardiovascular ischaemic event or type 2 diabetes. As for ischaemic events, we included myocardial infarction, stroke, or sudden death. Type 2 diabetes was defined as a fasting glucose ≥ 126 mg/dL or being treated with oral hypoglycaemic drugs or insulin. We analysed plasma and whole blood samples from the earliest samples available during 2021. For each case individually, we selected one control matched by age (±5 years), sex, smoking status (current/former, never), pre-ART CD4 cell count, viral load, and time of sample extraction (difference from cohort inclusion and sample extraction). If more than one control was available for one case, the matched control was selected randomly. Unfortunately, cardiovascular risk factors were not collected initially in this cohort.

### 2.2. Plasma Biomarkers

The stored plasma samples were tested for the following monocyte and macrophage activation biomarkers using specific ELISA kits (Bio-techne R&D Systems, Minneapolis, MN, USA): soluble CD14 (sCD14), soluble CD163 (sCD163), and monocyte chemoattractant protein-1 (CCL2/MCP-1), according to the manufacturer’s instructions. The case and control samples were assayed in duplicate and on the same ELISA plate.

### 2.3. Blood Telomere Length Measurement

According to the manufacturer’s protocol, genomic DNA was isolated from peripheral blood using the QIAamp DNA Blood Midi Kit (QIAGEN, Hilden, Germany). We used a NanoDrop^®^ ND-2000 spectrophotometer (Thermo Scientific, Waltham, MA, USA) for DNA quantification.

The relative telomere length, expressed as the ratio of telomere (T) to single-copy gen (S), was determined by monochrome quantitative multiplex PCR assay with minor modifications, as described in previous studies of our group [28]. A standard curve was prepared with genomic DNA from a pool of three healthy volunteers by serial dilution. We included a reference sample and negative control in triplicate in each run. The samples were assayed in triplicate on the same PCR plate, and those with a coefficient of variation (CV) greater than 10% were reanalysed.

### 2.4. DNA Methylation Analysis and Epigenetic Age Prediction

We used the Weidner ageing method to study epigenetic ageing in a blood sample based on the methylation levels of 3 CpGs located in three different genes (cg0228185: ASPA, cg25809905: ITGA2B, and cg17861230: PDE4C). To assess DNA methylation, genomic DNA was isolated from PBMC by QIAamp DNA blood midi kit (QIAGEN, Hilden, Germany; GYM) according to the manufacturer’s protocol. Subsequently, 1.5 μg of DNA per sample was bisulphite-converted using the EpiTect Bisulfite kit (QIAGEN, Hilden, Germany). PCR amplified the transformed DNA in the three genomic regions spanning the CpG of interest using modified primers based on Endo and colleagues’ design [29]. The PCR products were then analysed using Next Generation Sequencing (NGS) on a MiSeq sequencer (Illumina, San Diego, CA, USA).

For epigenetic age calculation, the methylation levels of the three CpGs were input into the Weidner ageing formula [30]:$$Predicted\;epigenetic\;age\left(in\;years\right)=38.0-26.4\alpha -23.7\beta +164.7\gamma$$

Epigenetic age acceleration was calculated as the residuals that result from regressing epigenetic age on chronological age; therefore, positive values imply that the epigenetic age is higher than predicted for the model for an individual of the same chronological age.

### 2.5. Statistical Analysis

The characteristics of the participants were described using absolute and relative frequencies and medians (interquartile range, IQR) for categorical and continuous variables, respectively. Conditional logistic regression analysis was used to study the association between cytokines, TL, and epigenetic age with cardiometabolic events and all combined events. Spearman rank correlations explored the associations between biomarker levels, TL, EAA, and clinical characteristics (chronological age, CD4+ cell count, and viral load).

All statistical analyses were performed using Stata software (version 14.0: Stata Corporation, College Station, TX, USA).

## 3. Results

### 3.1. Description of the Study Population

The analysis included 218 samples from 109 cases and 109 controls. We restricted this analysis to the 188 subjects with samples collected before the cardiometabolic event (Figure 1).

Most of the samples obtained were collected before ART initiation (126 samples; 67%). In addition, 62 collected after ART initiation (31 cases and 31 controls) matched for time sample extraction. We found 94 cases (19 myocardial infarctions, three sudden deaths, 12 strokes, and 60 type 2 diabetes). The cases and controls had similar characteristics (Table 1). Overall, 84% of the population were male, 84% had a sexually acquired infection, the median (IQR) age was 46 (40–56) years, the CD4+ T-cell count (cells/mm^3^) was 418 (227–653), and the plasma log HIV RNA viral load was 4.52 (IQR 3.77–5.09). The median TL (T/S ratio) was 1.09 (0.97–1.23), and EAA was −0.25 years (−3.51 to 3.95) years in the whole sample population.

### 3.2. Associations between Monocyte Biomarkers, TL, and EAA with Cardiometabolic Events

The monocyte biomarkers, TL, and EAA values between cases and controls are presented in Figure 2.

The relative telomere length was measured as the mean ratio of the telomere amplification product (T) to a single-copy gene (A). The boxes show median and IQR; the whiskers correspond to 10–90 percentiles. The *p*-value is from the conditional logistic regression model comparing distributions between cases and controls. All comparisons were non-significant.

No statistically significant differences existed between the cases and controls in any of the markers measured. The median (IQR) TL was 1.09 (0.94–1.23) in the cases and 1.10 (0.98–1.22) in the controls, and the median (IQR) EAA was 0.10 (−2.77 to 3.98) in the cases and −0.45 (−3.86 to 3.26) in the controls. The logistic regression analysis explored any association between the cardiometabolic event and ageing or monocyte biomarkers analysed as continuous variables or binary based on median categorisation. We found no statistically significant associations (Table 2).

These results did not change when the analysis was restricted to the 126 samples obtained before ART initiation. These results did not vary when the study of the pooled ischaemic events (myocardial infarction, sudden death, and stroke) and diabetes was run separately (Table 3 and Table 4).

### 3.3. Correlations of TL, EAA, and Monocyte Biomarkers

We evaluated the relationship between TL, EAA, and monocyte biomarkers (Figure 3). The monocyte/macrophage activation markers were weakly correlated with baseline viral load and CD4:CD8 ratio. The Spearman correlation coefficients (rho) with viral load for CCL2/MCP-1, sCD14, and sCD163 were −0.44 (*p* = 0.002), 0.15 (*p* = 0.049), and 0.17 (*p* = 0.028), respectively. sCD14 and sCD163 were negatively correlated with CD4:CD8 ratio (rho = −0.23; *p* = 0.001 and rho = −0.27; *p* = 0.002). There was also a weak positive correlation between chronological age and monocyte/macrophage activation markers sCD14 (rho = 0.21; *p* = 0.007) and sCD163 (rho = 0.17; *p* = 0.023). TL was negatively correlated with sCD14 (rho = −0.23; *p* = 0.01) and with both chronological (rho = −0.38; *p* < 0.001) and epigenetic age, as determined by the Weidner signature (rho = −0.19; *p* = 0.02), but not with EAA (rho = 0.13; *p* = 0.14). EAA was negatively correlated with CCL2/MCP-1 (rho = −0.17; *p* = 0.02).

## 4. Discussion

In this case–control study nested in the Spanish AIDS Research Cohort, we found no association between cardiometabolic events and TL, EAA, and monocyte inflammation markers.

Some studies have shown the association of TL with coronary heart disease and all-cause and cardiovascular-related mortality in the general population [31,32,33]. The relationship between TL and cardiovascular ageing is not clear enough, and other studies have demonstrated a lack of association between the leukocyte telomere attrition rate and atherosclerosis [34]. A recent review highlighted that most studies exploring the causal association between TL and atherosclerosis cardiovascular disease are cross-sectional and have shown a small effect size. It is still unknown whether telomere shortening is a cause or a consequence of cardiovascular disease [35]. Some meta-analyses support a weak relationship between leukocyte telomere shortening and cardiovascular disease and diabetes. One standard deviation decrease in TL was associated with a 21%, 24%, and 37% increased risk of stroke, myocardial infarction, and type 2 diabetes [16,36].

There is a shortage of studies exploring the association between TL and cardiovascular disease in PLHIV. In a recent case–control study including 1078 participants from the Swiss HIV cohort, Engel et al. demonstrated that participants with the 5th (longest) TL quintile had a 46% lower risk of coronary heart disease even after adjusting for classical cardiovascular and HIV-related risk factors [37]. To increase the number of events in our study, we decided to include diabetes as a metabolic event with a high risk for ischaemic events because it has been included previously in other studies [16]. The differences between these two studies could also be related to the smaller sample size of our research precluding enough statistical power, the lower number of ischaemic events (22 myocardial infarctions or sudden death in our study vs. 333 in the Swiss research), and the shorter time from sample to event time (2.4 years in our research and 9.4 years in Engel’s study). Apart from this, some other factors could have influenced our results. First, some samples in our study were obtained after ART initiation, which could affect the results. One study of 51 intravenous drug users showed that three months after HIV seroconversion, the TL in peripheral blood mononuclear cells decreased by 13%. It is well known that ART initiation positively impacts TL, reflecting the partial reversal of HIV-associated immunosenescence [38]. Second, but more importantly, the lack of data about cardiovascular risk factors in our cohort precludes us from ruling out residual confounding. The shorter median time to event in our study could reflect the effect of cardiovascular risk factors instead of TL, which could need a longer follow-up to influence cardiovascular health.

Additionally, PLHIV receiving long-term ART who maintain virological suppression continue experiencing mean blood TL gains ten years after achieving virological suppression [39,40]. This effect suggests that immune reconstitution and the reversal of immune senescence continue long after ART initiation. The fact that most cardiovascular risk factors also impact TL could have precluded us from finding an association between TL and cardiometabolic effect in this cohort.

Recent reports have identified different methylation loci associated with acute myocardial infarction; some were related to incident cases of coronary heart and cardiovascular disease. However, their utility as predictive biomarkers has not been demonstrated [41,42]. First-generation epigenetic clocks are excellent predictors of chronological age, even predicting time to death [18,19,43,44]. The second-generation epigenetic clocks have replaced chronological age with surrogate markers of a biological age-weighted average of biomarkers correlated with age (Phenoage) or levels of seven plasma proteins and CpGs associated with smoking (GrimAge) [45,46]. These clocks predict the time to coronary heart disease, mortality, and multimorbidity more accurately. We have used a minimised clock, based on 3CpGs, validated in blood samples to predict age. The correlation between inferred age from the Weidner formula with chronological age in the present study was good (rho 0.785; *p* < 0.001). Nevertheless, it cannot capture other aspects of ageing, such as comorbidities [30].

There are some concerns about methylation studies. DNA methylation patterns are highly tissue-specific. This is of utmost importance, especially for causative studies. Differences in cell subtype populations are also a significant confounder, and we did not control for this factor. A substantial limitation in cross-sectional studies is reverse causation when the disease produces changes in DNA methylation rather than vice versa. And this has been demonstrated in several studies showing how HIV infection produces a differential methylation pattern and accelerates epigenetic age even with high CD4 + cell count only partially restored after ART initiation [24,47,48].

Finally, although previous studies have demonstrated that single measurements of sCD14 and sCD163 are related to non-AIDS events, we did not find this association. Angelidou et al. showed that inflammation markers increased before a non-AIDS event, especially from year 1 to pre-event time [49]. We found no relationship between changes in monocyte inflammation biomarkers, TL, and EAA with ischaemic events [49]. This could be explained by the fact that we could not isolate the effect of ART on inflammation, as our samples were obtained during the first year of ART initiation. We only included cardiovascular ischaemic events in this analysis; in Angelidou’s study, there were more non-AIDS cancers and bacterial infections. To our knowledge, no studies show an association between telomere attrition and cardiovascular events, and in the Swiss cohort study, the TL measurement near the event was not helpful. Interestingly, in a recent survey of our group, we found a tendency for an association between EAA and some clinical events such as cardiovascular disease, diabetes, or cancer [50].

The main limitation of our study is the sample size, the low numbers of cardiovascular ischaemic events, and the short follow-up with a median time to events from cohort inclusion of 2.4 years. Most association studies between TL and cardiovascular events have shown little effect despite a large sample size. We could speculate that the sample size has more influence on the positive impact of those studies than the cardiovascular effect. Apart from these factors, we cannot avoid residual confounding with this case–control design, as many confounding factors were not recorded systematically in the original cohort (cardiovascular risk factors, ART drugs influencing methylation, and TL-related factors). The lack of samples after ART initiation when HIV-RNA is undetectable, and the lack of control for cell type in the DNA methylation analysis may also impact the results. Finally, the three-CpG Weidner clock, although sufficient to predict age accurately, may not be sensitive enough to capture phenotypic comorbidities such as cardiovascular disease or diabetes.

In conclusion, we did not find any associations between TL, EAA, and inflammatory cytokines with cardiovascular events and diabetes in this cohort. Further studies with larger sample sizes and longer follow-ups are needed to elucidate the role of these novel biomarkers of ageing in cardiovascular morbidity in PLHIV.

## Figures and Tables

**Figure 1 microorganisms-11-01818-f001:**
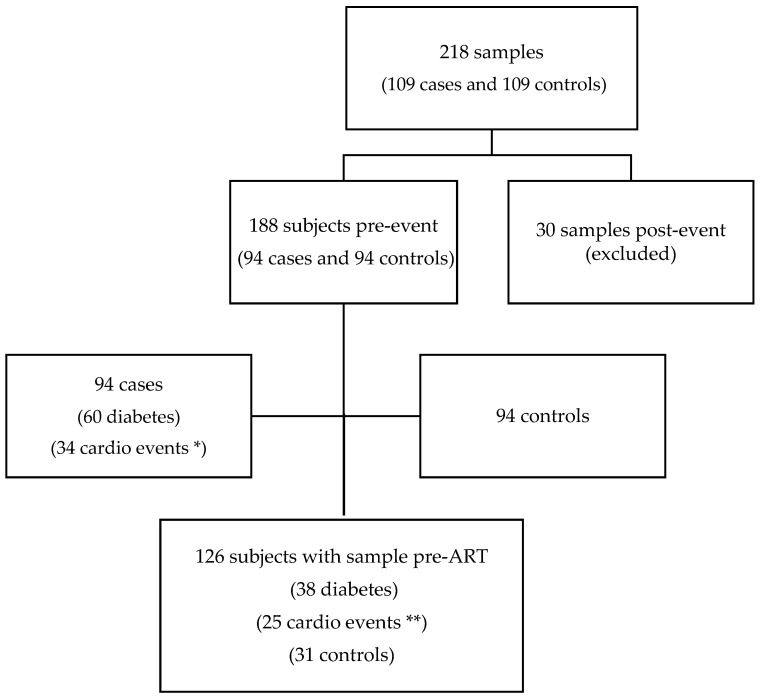
Flow chart of the study samples and participants. * 19 myocardial infarctions, 12 strokes, and three sudden deaths. ** 13 myocardial infarctions, nine strokes, and three sudden deaths. Sixty-two subjects (31 cases and 31 controls) had samples after antiretroviral treatment (ART) initiation.

**Figure 2 microorganisms-11-01818-f002:**
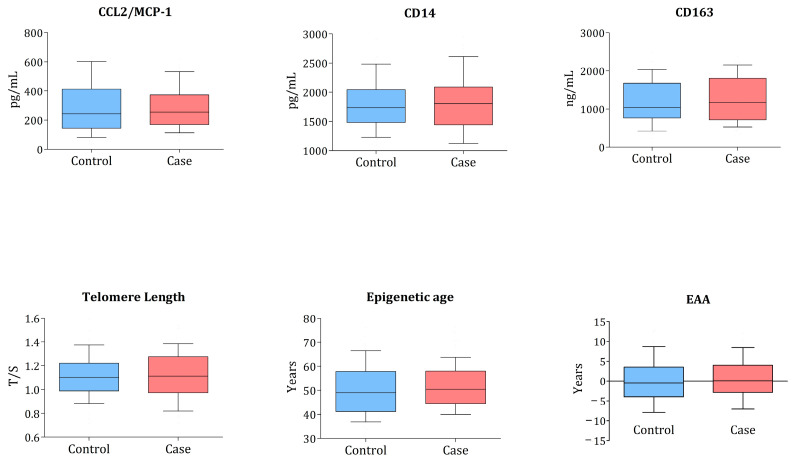
Box-and-whisker plots of monocyte activation markers, telomere length, and epigenetic age acceleration.

**Figure 3 microorganisms-11-01818-f003:**
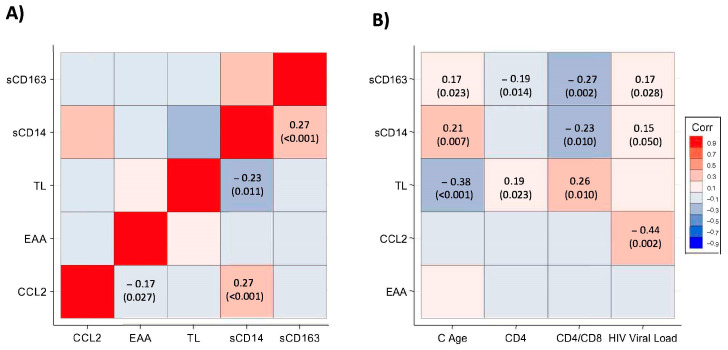
Correlations (*p*-values) of monocyte and ageing biomarkers (**A**) and with HIV-related variables (**B**). Negative correlations are in blue, and positive in red. Correlations used a Spearman rank test. TL: telomere length. EAA: epigenetic age acceleration, C Age: chronological age.

**Table 1 microorganisms-11-01818-t001:** Baseline characteristics of cases and controls (*n* = 188).

	Cases *n* = 94	Controls *n* = 94	*p*-Value
Age (years)	46 (39–56)	46 (40–55)	0.763
Female sex	15 (16.0%)	15 (16.0%)	
MSM	57 (60.6%)	45 (47.9%)	0.079
Caucasic	72 (76.6%)	84 (89.36%)	0.106
HIV duration	8.23 (1.51–26.16)	3.59 (1.25–18.30)	0.846
Never smoked	29 (30.9%)	29 (30.9%)	
CD4 (cells/mm^3^)	384 (234–574)	442 (226–692)	0.375
CD4 < 200 (%)	21 (22.3%)	21 (22.3%)	
Viral load (log)	4.53 (3.65–5.07)	4.49 (3.85–5.13)	0.847
CD4:CD8 ratio	0.33 (0.20–0.62)	0.44 (0.26–0.63)	0.292
Time to event (years)	2.39 (0.78–4.50)	------	
TL (T/S ratio)	1.09 (0.94–1.23)	1.10 (0.98–1.22)	0.965
Epigenetic age (years)	50.1 (43.8–57.2)	49 (42.6–56.6)	0.400
EAA (years)	0.10 (−2.7; 3.98)	−0.45 (−3.86; 3.26)	0.827
sCD14 (pg/mL)	1771.9 (1437.1–2105.5)	1751.5 (1497–2081.7)	0.591
sCD163 (pg/mL)	1192.4 (648.9–1903.1)	1022.4 (696.5–1586.61)	0.782
CCL2/MCP-1 (ng/mL)	260.2 (178.6–373.2)	261.4 (153.63–414.92)	0.520

Data are *n* (%) or median interquartile range (IQR). MSM: men who have sex with men. TL: telomere length.T/S ratio: telomere to single-copy gen. EAA: epigenetic age acceleration.

**Table 2 microorganisms-11-01818-t002:** Association between cardiometabolic events and LTL, age acceleration, and biomarkers.

	As Continuous Variable			As Binary Variable		
	Odds Ratio	95% CI	*p*-Value	Odds Ratio	95% CI	*p*-Value
TL (T/S ratio)	0.96	0.16–5.66	0.965	1.09	0.52–2.28	0.827
EAA (Years)	1.01	0.96–1.06	0.668	1.35	0.71–2.58	0.365
sCD14	1.0	1.0–1.0	0.591	1.29	0.66–2.54	0.456
sCD163	1.0	1.0–1.0	0.782	1.86	0.89–3.87	0.097
CCL2/MCP1	1.0	1.0–1.0	0.520	0.98	0.46–2.06	0.951

Data are *n* (%) or median interquartile range (IQR). TL: telomere length. T/S ratio: telomere to single-copy gen. EAA: epigenetic age acceleration. Odds ratio of the conditional logistic regression analysis for the presence of cardiometabolic events based on different biomarkers. Adjusted by time from blood sample to event and to end of follow-up. Binary variable based on median categorisation: TL (T/S ratio) > 1.09; EAA (years) > −0.25; sCD14 > 1757.24 pg/mL; sCD163 > 1084.23 pg/mL; CCL2/MCP1 > 261.45 ng/mL.

**Table 3 microorganisms-11-01818-t003:** Association between cardiometabolic events and LTL, age acceleration, and biomarkers in the subgroup of participants with ischaemic events (*n* = 68 (34 cases and 34 controls)).

	As Continuous Variable			As Binary Variable		
	Odds Ratio	95% CI	*p*-Value	Odds Ratio	95% CI	*p*-Value
TL (T/S ratio)	0.21	0.01–3.39	0.268	0.63	0.18–2.21	0.471
EAA (Years)	0.98	0.90–1.06	0.594	0.98	0.90–1.06	0.594
sCD14	1.0	1.0–1.0	0.149	1.45	0.47–4.45	0.512
sCD163	1.0	1.0–1.0	0.990	1.75	0.43–7.17	0.436
CCL2/MCP1	1.0	0.99–1.0	0.409	1.20	0.35–4.07	0.772

Data are *n* (%) or median interquartile range (IQR). TL: telomere length. T/S ratio: telomere to single-copy gen. EAA: epigenetic age acceleration. Odds ratio of the conditional logistic regression analysis for the presence of ischaemic events based on different biomarkers. Adjusted by time from blood sample to event and to end of follow-up. Binary variable based on median categorisation: TL (T/S ratio) > 1.09; EAA (years) > −0.25; sCD14 > 1757.24 pg/mL; sCD163 > 1084.23 pg/mL; CCL2/MCP1 > 261.45 ng/mL.

**Table 4 microorganisms-11-01818-t004:** Association between cardiometabolic events and LTL, age acceleration, and biomarkers in the subgroup of participants with diabetes (*n* = 120 (60 cases and 60 controls)).

	As Continuous Variable			As Binary Variable		
	Odds Ratio	95% CI	*p*-Value	Odds Ratio	95% CI	*p*-Value
TL (T/S ratio)	4.23	0.25–70.64	0.315	1.43	0.50–4.10	0.500
EAA (Years)	1.01	0.96–1.07	0.610	1.01	0.96–1.07	0.610
sCD14	1.0	1.0–1.0	0.578	1.07	0.44–2.59	0.878
sCD163	1.0	1.0–1.0	0.939	1.62	0.67–3.92	0.282
CCL2/MCP1	1.0	1.0–1.0	0.265	0.89	0.34–2.32	0.813

Data are *n* (%) or median interquartile range (IQR). TL: telomere length. T/S ratio: telomere to single-copy gen. EAA: epigenetic age acceleration. Odds ratio of the conditional logistic regression analysis for the presence of diabetes based on different biomarkers. Adjusted by time from blood sample to event and to end of follow-up. Binary variable based on median categorisation: TL (T/S ratio) > 1.09; EAA (years) > −0.25; sCD14 > 1757.24 pg/mL; sCD163 > 1084.23 pg/mL; CCL2/MCP1 > 261.45 ng/mL.

## Data Availability

The data presented in this study are available on request from the corresponding author. The data are not publicly available due to privacy restrictions.

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
