# Peer review of "Monocyte Activation and Ageing Biomarkers in the Development of Cardiovascular Ischaemic Events or Diabetes in People with HIV"

_microorganisms, 2023, doi:10.3390/microorganisms11071818_

Round 1
Reviewer 1 Report
Bernardino et al. have attempted to address an important clinical question of whether biomarkers can be identified that clinicians can use to identify people living with HIV (PLWH) who may be at higher risk of developing either diabetes or critical cardiovascular events than their peers. The biomarkers they chose to study, telomere length, epigenetic aging and soluble inflammatory monocyte cytokines, have been implicated in the pathogenesis of the events under study so these were valid and important markers to investigate. To address these questions, the authors undertook a nested case control study using samples from participants enrolled in the Spanish cohort of people living with HIV known as CoRIS. While they did find shorter telomeres and more advanced epigenetic aging in both cases and controls, the investigators did not find any statistically significant associations between the biomarkers measured and their chosen outcomes.
General Comments:
1) The study addresses an important topic that is of high importance to the field. The manuscript is clearly written and the data is well displayed. The investigators have drawn the appropriate conclusions from their data, while acknowledging the many limitations of the study design.
Major comments:
1) There were many limitations of this study which were well articulated in the discussion section. However, the accumulation of these limitations significantly limits the reliability of the conclusions.
2) One of the limitations is the use of Weidner’s measurement of epigenetic age acceleration which utilizes only 3 CpG sites and was constructed to correlate with chronologic age, not biologic age, unlike several of the second-generation clocks such as the Extrinsic, Phenotypic or GRIM age clocks. Genome wide methylation analysis would have allowed a more extensive analysis of the relationship between epigenetic aging and cardiometabolic outcomes, but it is admittedly rather expensive. Given this limitation, an alternate title should be considered as the phrase “Epigenetic Age Acceleration” in the title is rather too broad and could be misleading.
Specific Comments
1) On page 2, lines 87 and 88, the authors have written that “All centers are invited to provide data on the non-AIDS event,”. The word “invited” should be elaborated upon here. In the discussion it is noted that one of the limitations to the study is that “many confounding factors were not recorded systematically in the original cohort”. The word “invite” leaves the reader to wonder if all outcome events were adequately recorded. Is it possible that controls may have also experienced some of these outcomes but they were not required to collect and enter that data?
2) One page 6, line 204, the authors state “TL was negatively correlated with sCD14 (rho=‐0.23; p=0.01) and with both chronological (rho=‐0.38; p<0.001) and biological age (rho=‐0.19; p=0.02), but not with EAA (rho=0.13; p=0.14).” It was unclear what the difference is between biologic age and EAA. The initial impression is that EAA was used to determine biologic age, but this sentence implies otherwise.
3) Page 7, line 241, it may be more accurate for the authors to state “reflecting the partial reversal of HIV-associated immunosenescence. Partial more adequately describes the immune status of individuals on ART as many perturbations remain in the makeup of the immune system after ART initiation. The original wording “restoration” would imply that ART restores the immunosenescence elicited by ART, hence the suggestion to replace that word with “reversal”.
The English in the manuscript is very good and only a few minor edits would be suggested.
Author Response
Reviewer 1:
Major comments:
There were many limitations of this study which were well articulated in the discussion section. However, the accumulation of these limitations significantly limits the reliability of the conclusions.
We thank the reviewer for this comment. We also agree that the study have some limitations that we acknowledge and discuss thoroughly in the discussion section. Although our study is small, other studies with a larger sample size have also drawn conflicting results. We believe that this study could help researchers to optimize future projects and find a proper target population.
One of the limitations is the use of Weidner’s measurement of epigenetic age acceleration which utilizes only 3 CpG sites and was constructed to correlate with chronologic age, not biologic age, unlike several of the second-generation clocks such as the Extrinsic, Phenotypic or GRIM age clocks. Genome wide methylation analysis would have allowed a more extensive analysis of the relationship between epigenetic aging and cardiometabolic outcomes, but it is admittedly rather expensive. Given this limitation, an alternate title should be considered as the phrase “Epigenetic Age Acceleration” in the title is rather too broad and could be misleading.
We agree with the reviewer that using the second-generation epigenetic clocks would have been more appropriate to explore the association between epigenetic age acceleration and cardiometabolic events. Unfortunately, due to budget constrains in this project this could not be possible. Although others epigenetic clocks have better precision capturing age-related co-morbidities, in this study we tried to analyze if DNA-methylation age was related to cardiometabolic events in PLHIV. To this extent the Weidner formula could be useful as it is validated in blood samples and have an excellent correlation with chronological age. Nevertheless we accept the reviewer’s suggestion and have changed the title to: “Monocyte Activation and Aging Biomarkers in the development of Cardiovascular Ischemic Events or Diabetes in People with HIV“
Specific comments:
On page 2, lines 87 and 88, the authors have written that “All centers are invited to provide data on the non-AIDS event,”. The word “invited” should be elaborated upon here. In the discussion it is noted that one of the limitations to the study is that “many confounding factors were not recorded systematically in the original cohort”. The word “invite” leaves the reader to wonder if all outcome events were adequately recorded. Is it possible that controls may have also experienced some of these outcomes but they were not required to collect and enter that data?
We thank the reviewer for raising the lack of clarity. CoRis cohort is an open, prospective multicenter cohort of adults with confirmed HIV infection naïve to ART at entry since 2004. In 2008, the scientific committee of the cohort decided to start collecting data on non-AIDS clinical events (please see reference 23 for more information). All participating centres provided data on cardiovascular clinical endpoints and diabetes from 2008 onwards. All these clinical events are recorded in the cohort in a structured reporting form. We have amended the wording and changed “All centres are invited to provide data on the incident non-AIDS event…” for “All centres provided data on the incident non-AIDS-event…”
On page 6, line 204, the authors state “TL was negatively correlated with sCD14 (rho=‐0.23; p=0.01) and with both chronological (rho=‐0.38; p<0.001) and biological age (rho=‐0.19; p=0.02), but not with EAA (rho=0.13; p=0.14).” It was unclear what the difference is between biologic age and EAA. The initial impression is that EAA was used to determine biologic age, but this sentence implies otherwise.
We thank the reviewer for this comment. We agree that the terminology biologic age and epigenetic age acceleration may be confusing, but we think the difference in these terms is explained in the methods section.
“Epigenetic age is the predicted age resulting of the Weidner aging formula. Epigenetic age acceleration is the residual resulting from the linear regression (biologic age on chronological age).” Therefore, epigenetic age acceleration is the deviation of the epigenetic age from the predicted epigenetic age based on chronological age. Positives values imply a faster epigenetic age acceleration (epigenetic age older than predicted epigenetic age). Nevertheless, we have substituted the term biological age for epigenetic age in this paragraph.
Page 7, line 241, it may be more accurate for the authors to state “reflecting the partial reversal of HIV-associated immunosenescence. Partial more adequately describes the immune status of individuals on ART as many perturbations remain in the makeup of the immune system after ART initiation. The original wording “restoration” would imply that ART restores the immunosenescence elicited by ART, hence the suggestion to replace that word with “reversal”.
We agree with the reviewer and thanks for this comment. We have corrected the sentence in the manuscript as suggested.
Comments on the Quality of English Language
The English in the manuscript is very good and only a few minor edits would be suggested.
Thank you for the comment. We have reviewed the manuscript and minor editing have been set.
Reviewer 2 Report
General comments:
In this paper, the authors utilized a case-control design to investigate differences in inflammation and aging-related biomarkers between people with HIV (PWH) who experienced a cardiovascular disease (CVD) event versus those who did not. The authors ultimately found null results that could be largely attributed to the small sample sizes. The introduction and discussion sections were well written. The reporting of methods and results can be improved. See detailed comments below.
Detailed comments:
1. The reasoning behind combining Type 2 diabetes and CVD event for identifying cases is unclear to me. While the conditions share common pathways, it is probably unwise to combine the two. Citing that other studies did it is not a good enough reason.
2. The phrasing in the methodology section could be improved to clarify that samples included in the analysis were collected prior to the event. I had to double-check with the results section to make sure that I understood when the samples were collected. Figure 1 can be updated to make it clear what was included in the final analytic sample. They should be explicit in how they conducted the matching (random? measure of closeness? handling of multiple matches?).
3. The authors report dropping 30 individuals due to missing samples. I would like to see some discussion on how this missing data issue could have affected their results. The authors could also conduct bias analyses to see how results may have changed in the scenario that biomarker data was available for this group.
4. The authors need to clarify the “time of sample extraction”. It is unclear if it is just the calendar date or time relative to the event or time relative to cohort entry. This mismatch in timing of samples could also explain why their results are null.
5. Because of how they ordered the matching and exclusions, the analytic sample is a selected sample from the original sample. They need to discuss how selection bias can influence their findings and clarify what population they are generalizing to with their results. For example, they conclude “we did not find any associations… in this cohort”. It is unclear which cohort they are referring to: the whole Spanish HIV cohort vs their original matched cohort vs their final analytic cohort.
6. The authors need to show the results mentioned in lines 193 and 194.
7. In Figure 2, they need to report what “+” means.
8. In Table 2, they should put in the footnote how the binary categorization was done. They should also mention in the footnote any additional variables they included in the logistic model.
English is fine. Minor typos that needs to be checked prior to final publication.
Author Response
Major comments:
In this paper, the authors utilized a case-control design to investigate differences in inflammation and aging-related biomarkers between people with HIV (PWH) who experienced a cardiovascular disease (CVD) event versus those who did not. The authors ultimately found null results that could be largely attributed to the small sample sizes. The introduction and discussion sections were well written. The reporting of methods and results can be improved. See detailed comments below.
The reasoning behind combining Type 2 diabetes and CVD event for identifying cases is unclear to me. While the conditions share common pathways, it is probably unwise to combine the two. Citing that other studies did it is not a good enough reason.
We appreciate the reviewer for this important comment. The Spanish CoRIS cohort includes adult PWH naïve to antiretroviral treatment. Although this cohort started inclusions in 2004, the participants with available samples in the repository biobank were included in the cohort more recently. Therefore, the chance for a cardiovascular endpoint was low. The numbers of myocardial infarction, stroke and sudden death in this cohort are low, and the median age of participants is younger than other studies about cardiovascular endpoints (46 years). Therefore, the reasons for the composite outcome including diabetes as an endpoint were the following: the small sample size in our study with few hard ischaemic endpoints (myocardial infarction, stroke or sudden death), the higher cardiovascular risk of diabetic persons and the thoughts that diabetes is a coronary equivalent (in terms of cardiovascular risk), and the fact that previous studies have shown a small effect of telomere length and CVD, and diabetes in large cohorts in the general population. Several studies in the literature analyze the association of telomeres length with diabetes and CV events together (metanalyses of D’Mello Circ Cardiovasc Genet 2015 and Haycock BMJ 2014). Recent data demonstrate the potential use of DNA methylation as a potential risk factor for type 2 diabetes development (Kim K. Diabetes 2021;70:1404-1413. PMID: 33820761
The phrasing in the methodology section could be improved to clarify that samples included in the analysis were collected prior to the event. I had to double-check with the results section to make sure that I understood when the samples were collected. Figure 1 can be updated to make it clear what was included in the final analytic sample. They should be explicit in how they conducted the matching (random? measure of closeness? handling of multiple matches?).
We thank the reviewer for this comment. We have added the following information regarding the matching in methods section: LINE 355-356. “We requested for the first available sample preferentially before ART initiation”. We have also amended the figure in order to clarify the numbers. We have added information regarding the matching in methods sections: LINE 362-366. “For each case individually, we selected one control matched by age (± 5 years), sex, smoking status (current/former, never), pre-ART CD4 cell count, viral load, and time of sample extraction (difference from cohort inclusion and sample extraction). If more than one control was available for one case, the matched control was selected randomly”.
The authors report dropping 30 individuals due to missing samples. I would like to see some discussion on how this missing data issue could have affected their results. The authors could also conduct bias analyses to see how results may have changed in the scenario that biomarker data was available for this group.
We would like to clarify that we did not lose any individual from the analysis. This is a misunderstanding as we probably have not been clear enough. There were 30 blood samples that the collection date was after the cardio or diabetes event, therefore we excluded this samples from the analysis.
The authors need to clarify the “time of sample extraction”. It is unclear if it is just the calendar date or time relative to the event or time relative to cohort entry. This mismatch in timing of samples could also explain why their results are null.
Thank you very much for this comment and your critical review of the manuscript. In this cohort investigators are requested to extract at least yearly samples for biobank. Unfortunately, this is not always the rule, and since the cohort was established in 2004 there are some individuals with no samples available as have been used in other projects. We define time of sample extraction as the time relative to cohort entry as we thought cases and controls should have the same period of observation within the cohort. We agree with the reviewer that this information is crucial to understand the findings. We have added this information in the methods LINES 364-365: (difference from cohort inclusion and sample extraction)
Because of how they ordered the matching and exclusions, the analytic sample is a selected sample from the original sample. They need to discuss how selection bias can influence their findings and clarify what population they are generalizing to with their results. For example, they conclude “we did not find any associations… in this cohort”. It is unclear which cohort they are referring to: the whole Spanish HIV cohort vs their original matched cohort vs their final analytic cohort.
Thank you for the comment. In our case control study, we included all cases in the cohort with available blood sample and we selected individually matched controls. We think that the selection of controls is not affected by selection bias because within the constraints of any matching criteria, the exposure to risk factors and confounders is representative of that in the population “at risk” for ischemic events and diabetes. The controls are subjects extracted from the same population than cases who do not have cardiovascular events or DM, but who would be included in the study as cases if they had had the criteria.
Selection bias could have been introduced by selecting cases and controls with available blood samples therefore our results could be generalized to the Spanish HIV cohort with available blood samples in the biobank.
The authors need to show the results mentioned in lines 193 and 194.
We agree with the reviewer that this information could be useful for readers. We have added additional tables as supplementary information with this analysis.
In Figure 2, they need to report what “+” means.
Figure 2 have been corrected. We think the new version is clearer.
In Table 2, they should put in the footnote how the binary categorization was done. They should also mention in the footnote any additional variables they included in the logistic model.
We have added the requested information in the table and in the footnote
Comments on the Quality of English Language
The English in the manuscript is very good and only a few minor edits would be suggested.
Thank you for the comment. We have reviewed the manuscript and minor editing have been set.
Reviewer 3 Report
The manuscript entitled, “Monocyte Activation, Telomere Length, Epigenetic Age Acceleration and Development of Cardiovascular Ischemic Events or Diabetes in People with HIV" by Bernardino et al., examined if in HIV-positive individuals, blood soluble inflammatory monocyte cytokines, blood telomere length, and epigenetic age acceleration are related to cardiovascular events or diabetes. The Weidner method was used for the analysis of TL (T/S ratio) utilizing qPCR, EAA, and DNA methylation alterations discovered using next-generation sequencing. To investigate the relationship with cardio-metabolic events, authors employed conditional logistic regression. 94 cases (22 myocardial infarction/sudden death, 12 strokes, and 60 DM) and 94 controls totaled 188 individuals participated in the study. 84% of participants were men, with a median (IQR) age of 46 (40-56), 53% of them currently smoking, 22% having a CD4 count below 200 cells/mm3, and a median (IQR) log viral load of 4.52 (3.77-5.09). Monocyte cytokines, EAA, and TL did not significantly predict cardiometabolic events in this study. Additionally, no links between TL, EAA, and monocyte cytokines and diabetes or cardiovascular events could be found.
While I agree that this is a necessary and appropriate study, I have some suggestions for improving the article so that it provides a more thorough analysis of the topic.
Comments:
1 The English of manuscript can be polished (minor), there are typological mistakes.
2, At least one illustrative figure may be provided as to highlight the summary of this study.
3, The author should cross-check all abbreviations in the manuscript. Initially, define in full name followed by abbreviation.
4, Role of immune cells is also very important factor in cardiovascular ischemic events, therefore I would suggest adding few citations to put comprehensive view of this topic (PMID: 36093172; PMID: 27031276; PMID: 34389841; PMID: 36337927; PMID: 30653442 etc.)
5, Figures 2 quality may be improved (high resolution).
6, Author should include statistical methods used in the figure legends.
Minor editing of English language required
Author Response
Major comments:
At least one illustrative figure may be provided as to highlight the summary of this study
We thank the reviewer for this comment and agree that graphical abstracts are useful to better understand the manuscript. Unfortunately, our project gives negative results, so it is quite difficult to summarize the findings in a figure. Nevertheless, we provide one figure with the study hypothesis for the reviewer and editors consideration.
The author should cross-check all abbreviations in the manuscript. Initially, define in full name followed by abbreviation.
Thanks for this comment. We have double checked all the abbreviations throughout the manuscript.
Role of immune cells is also very important factor in cardiovascular ischemic events, therefore I would suggest adding few citations to put comprehensive view of this topic
Again, thank you for the suggestion. As requested, we have added the following sentence in the introduction with the corresponding references LINES 165-166: “T-cell activation is a hallmark of atherosclerosis, and both innate and adaptive immune cells are key players in plaque formation and progression.” We have added two new references (11 and 12) as suggested.
Figure 2 quality may be improved (high resolution)
Thank you for the comment. We have improved figure quality.
When the article is visualized with a less than 110% zoom it looks a bit blurred, but when you increase the zoom to 200% the quality is quite good. I do not know if this problem could be solved in the editing process.
Author should include statistical methods used in the figure legends
The statistical methods have been added in the footnotes of the figures.
Comments on the Quality of English Language
The English in the manuscript is very good and only a few minor edits would be suggested.
Thank you for the comment. We have reviewed the manuscript and minor editing have been set.
Round 2
Reviewer 1 Report
Save one comment, the authors have addressed the specific issues previously raised. One comment remains to be clarified.
The previous comment read:
“One page 6, line 204, the authors state “TL was negatively correlated with sCD14 (rho=‐0.23; p=0.01) and with both chronological (rho=‐0.38; p<0.001) and biological age (rho=‐0.19; p=0.02), but not with EAA (rho=0.13; p=0.14).”
The authors have addressed this by replacing the word “biologic” with “epigenetic”. Epigenetic age is a better term, but to fully remove confusion, perhaps they should simply state "epigenetic age, as determined by the Weidner signature, but not with EAA."
The largest concern with the previous version of the paper was the high number of limitations with the study design which significantly limited interpretation of the results. That has been moderately mitigated by clarifying that all sites provided data on incident non-AIDS events.
There are very minor concerns with the English. By addressing the few previous comments, the authors have enhanced the clarity of the article.
Author Response
Thank you for your comment, and sorry for not being clear enought. We accept your suggestion and have ammended the paragraph accordingly. See LINES 532-533 of the new version.